# Recurrent Implantation Failure: Bioinformatic Discovery of Biomarkers and Identification of Metabolic Subtypes

**DOI:** 10.3390/ijms241713488

**Published:** 2023-08-30

**Authors:** Yuan Fan, Cheng Shi, Nannan Huang, Fang Fang, Li Tian, Jianliu Wang

**Affiliations:** 1Department of Obstetrics and Gynecology, Peking University People’s Hospital, Beijing 100044, China; fanyuan94@bjmu.edu.cn (Y.F.); chengshi@bjmu.edu.cn (C.S.); huangnannan@pkuph.edu.cn (N.H.); fontana@bjmu.edu.cn (F.F.); 2Reproductive Medical Center, Peking University People’s Hospital, Beijing 100044, China

**Keywords:** recurrent implantation failure, metabolism-related genes, metabolic subtypes, immune infiltration

## Abstract

Recurrent implantation failure (RIF) is a challenging scenario from different standpoints. This study aimed to investigate its correlation with the endometrial metabolic characteristics. Transcriptomics data of 70 RIF and 99 normal endometrium tissues were retrieved from the Gene Expression Omnibus database. Common differentially expressed metabolism-related genes were extracted and various enrichment analyses were applied. Then, RIF was classified using a consensus clustering approach. Three machine learning methods were employed for screening key genes, and they were validated through the RT-qPCR experiment in the endometrium of 10 RIF and 10 healthy individuals. Receiver operator characteristic (ROC) curves were generated and validated by 20 RIF and 20 healthy individuals from Peking University People’s Hospital. We uncovered 109 RIF-related metabolic genes and proposed a novel two-subtype RIF classification according to their metabolic features. Eight characteristic genes (*SRD5A1*, *POLR3E*, *PPA2*, *PAPSS1*, *PRUNE*, *CA12*, *PDE6D*, and *RBKS*) were identified, and the area under curve (AUC) was 0.902 and the external validated AUC was 0.867. Higher immune cell infiltration levels were found in RIF patients and a metabolism-related regulatory network was constructed. Our work has explored the metabolic and immune characteristics of RIF, which paves a new road to future investigation of the related pathogenic mechanisms.

## 1. Introduction

Embryo implantation is a delicate and tightly regulated process, achieved by a synchronized and coordinated crosstalk between the embryo and the endometrium [1]. In recent decades, despite the tremendous advances in reproductive medicine, recurrent implantation failure (RIF) is still a controversial and poorly understood clinical problem. RIF affects about 10% worldwide patients undergoing in vitro fertilization and embryo transfer, being a challenge and a setback for both patients and clinicians [2]. RIF etiology is complex and is usually grouped into three categories involving the receptivity of the endometrium, the embryo, and their interacting environment. Within assisted reproductive technologies in which good-quality embryos are implanted, insufficient endometrial receptivity has resulted in approximately two-thirds of all implantation failures [3]. Thus, identifying molecular markers and clarifying the mechanisms is a strategy with important theoretical and clinical value.

The dynamic endometrial changes during the menstrual cycle are metabolically demanding. Aerobic glycolysis and lactate accumulation are important additional metabolic requirements of the implantation process [4,5,6]. During early pregnancy, the lactated uterine microenvironment seems to be favorable to embryo implantation [7]. During the decidualization process, genes and other factors related to aerobic glycolysis are extensively induced. On the other hand, the inhibition of lactate production can lead to decidua damage [8]. In addition, hyperinsulinemia and insulin resistance have been proven to inhibit the expression of endometrial receptivity markers, such as the insulin-like growth factor type 1 receptor [9]. Clinically, the low fertility of patients with metabolic diseases, such as polycystic ovary syndrome and diabetes, suggests that metabolic imbalance may affect endometrial receptivity and embryo implantation [10]. Nevertheless, the metabolic characteristics related to endometrial receptivity in RIF patients, and their potential influencing mechanisms and regulatory pathways, are still unclear.

Embryo invasion also requires specific immune activation at the maternal-fetal interface [11]. There are several types of immune cells in the endometrium, such as natural killer cells, macrophages, dendritic cells, and T cells. These cells are essential in regulating endometrial receptivity and embryo implantation [12]. Local immune dysfunction can damage endometrial receptivity and lead to RIF. Moreover, impairments in the maternal immune system during pregnancy can improve the susceptibility to viral infections, ultimately leading to an impairment of the embryo formation [13]. Several comprehensive bioinformatic analyses have indicated differences in immune cell infiltration levels between RIF patients and healthy individuals [14]. However, the mechanisms underlying immune cell infiltration in RIF patients are still worthy of further exploration.

In recent years, the rapid development of sequencing technology has provided us with a novel view of gene and protein expression patterns that could help to understand the mechanisms of implantation failure from different aspects [15,16,17]. However, most of these studies have a small sample size and have not been validated through an independent cohort. Moreover, the characteristics and mechanisms related to their metabolic component were not comprehensively analyzed in these studies. In view of the RIF complexity, it is necessary to explore its characteristics and pathogenesis from different perspectives, with the help of the rapidly developing high-throughput sequencing technology. This alternative approach can help to seek more efficient treatment strategies for RIF patients.

In the present study, we aimed to accurately explore metabolism-related genes related to endometrial receptivity, which enables the prediction of RIF occurrence from metabolic features. Moreover, a novel RIF classification, containing two subtypes with different metabolic characteristics, is proposed. Furthermore, we also investigated the immune infiltration and constructed the microRNA (miRNA)-transcription factor (TF)-genes network to gain a better understanding of the potential molecular metabolic process related to RIF occurrence. 

## 2. Results

### 2.1. Identification of Differentially Expressed Genes and Functional Enrichment Analysis

An overview of the workflow is shown in Figure 1. Initially, we combined the expression profiles of 70 RIF and 99 normal endometrium tissues from the GSE58144, GSE103465 and GSE111974 datasets cohorts. Before removing batch effects, endometrium tissues from different platforms showed significantly different clustering patterns but grouped together after batch correlation (Figure 2A,B and Appendix A). According to the predefined cut-off criteria, we detected a total of 1185 differentially expressed genes (DEGs), including 619 downregulated genes and 566 upregulated genes between the two groups of endometrium tissues among 17407 genes, as presented in Figure 2C,D. Gene Ontology (GO) enrichment analysis revealed that DEGs were enriched in regulation of mRNA metabolic process, histone modification, phosphoprotein phosphatase activity and so on (Figure 2E and Appendix A). In addition, the results of Kyoto Encyclopedia of Genes and Genomes (KEGG) pathway analysis indicated that DEGs were enriched in ubiquitin mediated proteolysis, glycerophospholipid metabolism, protein processing in endoplasmic reticulum and other functions (Figure 2F). Moreover, the GO and KEGG enrichment analyses of separated upregulated and downregulated RIF-related DEGs are shown in Appendix A. These results demonstrated that these DEGs were involved in various metabolic processes. 

### 2.2. Expression of Metabolism-Related Genes in Recurrent Implantation Failure Patients

We overlapped 1660 metabolic genes with the DEGs from the three merged datasets. The Venn diagram analysis revealed 109 overlapped metabolism-related genes, which were selected for further analysis. Of these, 54 were upregulated and 55 were downregulated (Figure 3A,B). To analyze the overall expression levels of metabolism-related genes, a volcano plot (Appendix A) and heatmap (Appendix A) of the expression levels of these metabolism-related genes were constructed. Through the GO and KEGG functional analyses, these metabolism-related genes were mainly linked to glycolipid metabolic processes, such as the phospholipid, glycerolipid, glycerophospholipid, and cholesterol metabolism, as well as fatty acid metabolism and biosynthesis, glycolysis/gluconeogenesis, and other metabolic pathways (Figure 3C–E, Appendix A). 

### 2.3. Construction and Characterization of Two Metabolic Subtype Models of Recurrent Implantation Failure 

Through the consensus clustering approach, RIFs were clustered in accordance with expression profiling of 109 metabolism-related genes. The optimal clustering stability was identified when K = 2, which was determined using a consensus matrix plot, a cumulative distribution function (CDF) plot, relative alterations in the area under the CDF curve, and a tracking plot (Figure 4A and Appendix A). The two metabolic subtypes were termed subtype-A and subtype-B, including 31 and 39 patients, respectively. Principal component analysis (PCA) revealed a remarkable difference between the two subtypes (Figure 4B). The heatmap and boxplot revealed a notable heterogeneity in the expression levels of these metabolism-related genes between two RIF subtypes (Figure 4C,D).

Gene set variation analysis (GSVA) was further conducted to explore any differences in metabolic pathway enrichment between the two subgroups. As shown in Figure 4E,F, the subtype A group was enriched in inflammasomes, inflammatory response, and adhesion molecules. The subtype B group was enriched in the biosynthesis of unsaturated fatty acids, fatty acid metabolism, mitochondrial fatty acid beta-oxidation, and cholesterol biosynthesis and homeostasis. While subtype A might have connections with inflammation pathways, subtype B was more closely associated with lipid metabolism.

DEGs were identified between the two clusters. To explore the underlying signaling mechanisms, functional analyses were performed. A total of 296 DEGs were detected, from which 98 genes were downregulated and 198 genes were upregulated in cluster 2, as compared with cluster 1 (Figure 5A). GO enrichment analysis showed that the DEGs were enriched in the hormone metabolic process, alcohol metabolic process, peptidase regulator activity, and endopeptidase regulator activity (Figure 5B). KEGG analysis showed that the DEGs were enriched in arachidonic acid metabolism, retinol metabolism, and linoleic acid metabolism (Figure 5C).

### 2.4. Selection of Characteristic Genes by Machine Learning Methods

Three algorithms were utilized for screening characteristic genes among key metabolism-related genes. For the least absolute shrinkage and selection operator (LASSO) algorithm, we selected the minimum criteria for building the LASSO classifier due to higher accuracy by comparisons and 40 characteristic genes were identified (Figure 6A,B). For the random forest algorithm, the top 10 characteristic genes with relative importance > 0.5 were determined (Figure 6C,D). For the support vector machine-recursive feature elimination (SVM-RFE) algorithm, when the feature number was 28, the classifier had the minimum error (Figure 6E). Following the intersection, 8 characteristic genes shared by the three algorithms were finally identified (*SRD5A1*, *POLR3E*, *PPA2*, *PAPSS1*, *PRUNE*, *CA12*, *PDE6D*, and *RBKS*; Figure 6F). Compared with the control group, the expression of *PAPSS1* was upregulated in the RIF group, while the other genes were downregulated.

### 2.5. Characteristics of Metabolism-Related Hub Genes

The analysis revealed a positive correlation between the expression levels of *SRD5A1*, *POLR3E*, *PPA2*, *PRUNE*, *CA12*, *PDE6D*, and *RBKS*. *PAPSS1* showed a significant negative correlation with *SRD5A1*, *POLR3E*, *CA12*, and *RBKS*. Among these genes, the strongest correlation existed between *PAPSS1* and *RBKS* (Figure 7A). Correlation analysis between the eight characteristic genes and all genes from the three datasets was carried out. The 50 genes with the strongest correlation were displayed in a heatmap (Appendix A). Based on the correlation analysis results, the gene set enrichment analysis (GSEA) of the single gene based on KEGG was prosecuted to evaluate signaling pathways involved in the characteristic genes (Figure 7B–I). Among them, *PAPSS1* was mainly involved in the metabolic pathways and amino sugar and nucleotide sugar metabolism. *PRUNE*, *PPA2* and *CA12* were mainly involved in the metabolic pathways, carbon metabolism and fatty acid metabolism. *RBKS* was mainly involved in the carbon metabolism and propanoate metabolism. *SRD5A1* was mainly involved in the glycolysis/gluconeogenesis, steroid hormone biosynthesis and carbon metabolism. 

### 2.6. Immunological Infiltration Features of Recurrent Implantation Failure

Immunological features were evaluated through the assessment of immune cell infiltration. There were also some interactions between immune cell populations across RIF (Figure 8A). Compared with the control group, more activated B cells, mast cells, and monocytes were found in RIF patients presenting higher immune infiltration levels (Figure 8B). Moreover, as illustrated in Figure 8C, the analyses displayed positive interactions between characteristic genes and immune cell infiltrations. The most strongly associated pairs included *SRD5A1* and immature B cell (r = 0.26, *p* < 0.001), *POLR3E* and type 17 T helper (Th17) cell (r = −0.42, *p* < 0.001), *PPA2* and neutrophil (r = −0.41, *p* < 0.001), *PRUNE* and CD56bright natural killer (NK) cell (r = −0.24, *p* = 0.002), *CA12* and T follicular helper cell (r = 0.39, *p* < 0.001), *PDE6D* and CD56bright NK cell (r = −0.44, *p* < 0.001), *RBKS* and CD56dim NK cell (r = 0.47, *p* < 0.001), *PAPSS1* and type 1 T helper (Th1) cell (r = −0.47, *p* < 0.001). Hence, the characteristic genes might modulate immunological features during the occurrence of RIF.

### 2.7. Diagnostic Efficacy and Validation of Characteristic Genes for Recurrent Implantation Failure Prediction

In the three combined cohorts, the diagnostic performance of each characteristic gene in RIF prediction was evaluated. The AUC values of the receiver operator characteristic (ROC) curves are shown in Figure 9A, indicating that these characteristic genes enabled to estimate the occurrence of RIF. When all of them were fitted into one variable, the area under curves (AUC) of the ROC curve was 0.902. Then, the diagnostic performance of the characteristic genes was validated by our patient cohort. Clinical characteristics of the recruited healthy individuals and RIF patients are presented in Table 1. The AUC was 0.867, which indicates that these eight genes were also potentially diagnostic markers for RIF (Figure 9B).

Next, to predict the prevalence of RIF in patients, a diagnostic nomogram was created based on the eight characteristic genes (Figure 9C). The calibration curves revealed that the line graph model predictions were nearly identical to those of the ideal model (Figure 9D). In addition, the single predicted risk score in the decision curve analysis or the composite genetic model was better than that in the random model. These results imply that decision-making based on the line graph model could be beneficial for RIF patients (Figure 9E).

### 2.8. Verification of Characteristic Genes

In order to experimentally validate the bioinformatics results, real time-quantitative PCR (RT-qPCR) experiments were performed to compare the expression levels of eight characteristic genes in the endometrium of 10 RIF patients and 10 healthy individuals. The results revealed that, in comparison with the healthy individuals, the mRNA levels of *SRD5A1*, *POLR3E*, *PPA2*, *PRUNE*, *CA12*, and *RBKS* were significantly lower, and the mRNA expression levels of *PAPSS1* were significantly higher in the RIF group. Moreover, the mRNA levels of *PDE6D* in the RIF group were lower than in the control group; however, there was no statistically significant difference between the two groups. This observation indicated that the data mining results are reliable and have further research value (Figure 10).

### 2.9. Establishment of the microRNA-Transcription Factor-Genes Network

To investigate the associated molecular mechanism, the RegNetwork database was utilized to identify upstream miRNAs and TFs of the eight target genes. Moreover, their interaction pairs (potentially involved in RIF regulation) were retrieved, to generate the regulatory interaction network (Figure 11). In this network, hsa-miR-134 was confirmed as a coregulator of *PAPSS1* and *POLR3E*, hsa-miR-206 was confirmed as a coregulator of *PAPSS1* and *PDE6D*, and hsa-miR-485-5p was confirmed as a coregulator of *PRUNE* and *PDE6D*. Furthermore, hsa-miR-580, hsa-miR-30a, and hsa-miR-30b were confirmed as coregulators of *PDE6D* and *POLR3E*. In addition, the TFs of MYC and YY1 were identified as a coregulator of *SRD5A1*, *POLR3E*, and *PDE6D*; the TFs of MAX and USF1 were proven to be coregulators of *SRD5A1* and *POLR3E*, and the TF of EGR1 was found to be a coregulator of *SRD5A1*. In addition, *RBKS* and *PAPSS1*, the TFs of HNF4A and BPTF were confirmed to be coregulators of *RBKS* and *PRUNE*, and the TFs of REST and SP1 were found to be coregulators of *PRUNE* and *POLR3E*.

## 3. Discussion

In infertile couples, RIF is a highly frustrating and distressing reproductive problem. Understanding its pathogenesis is helpful for its treatment and for improving patient outcomes [18]. Timely implantation of the blastocyst into the uterine endometrium is essential for the initiation of pregnancy. During early pregnancy in mammals, metabolic regulation plays important roles in embryo development and uterine receptivity, ultimately influencing pregnancy efficiency [4,5,6,7,8,19]. Therefore, uncovering metabolic characteristics related to endometrial receptivity and exploring potential druggable mechanisms can have a profound impact on improving the outcome of RIF patients. 

Nowadays, the etiology, effective diagnostic markers, and therapeutic strategies of RIF remain to be fully elucidated. In this study, we studied the pathological mechanisms of RIF from a new perspective. Furthermore, we explored a new strategy for the diagnosis of RIF, based on bioinformatic analysis combined with machine learning. Overall, we probed the involvement of metabolism in the pathological mechanisms of RIF. We revealed the metabolic characteristics in the endometrium during the days 20–24 in the luteal phase of the menstrual cycle: (1) Through differential expression analysis, we obtained 109 metabolism-related genes in RIF and investigated their specific molecular mechanisms through functional enrichment analysis. (2) We proposed a novel RIF classification, containing two subtypes, according to their metabolic features, and analyzed their correlation with immune infiltration levels. With this approach, we gain a better understanding of the potential molecular metabolic processes during the occurrence of RIF. (3) We implemented three machine learning algorithms to accurately explore the metabolic genes related to endometrial receptivity, including *SRD5A1*, *POLR3E*, *PPA2*, *PAPSS1*, *PRUNE*, *CA12*, *PDE6D,* and *RBKS*, which enabled the identification of RIF occurrence through metabolic characteristics. (4) Then, the diagnostic value of the eight gene expression signatures was evaluated by ROC analysis. This model was validated by an external clinical patient cohort from Peking University People’s Hospital (PKUPH). In addition, the eight key genes involved in RIF were validated in the endometrium samples of the enrolled patients. (5) Higher immune cell infiltration levels were found in RIF, which were positively linked to the characteristic genes. Our study provided some new insights into the potential pathogenesis of RIF and future research direction in this field.

To establish endometrial receptivity, a large amount of energy metabolism is required for the adaptation to changes in endometrial morphology, regulation of the endometrial environment and function, and the preparation for embryo implantation [20]. Among them, glucose and lipid metabolism play an important role in cellular energy and material sources [21]. In the present study, the functional enrichment analysis has uncovered 1185 DEGs between the RIF group and the control group. In addition, 109 overlapped metabolism-related genes were mainly involved in specific glucose and lipid metabolism processes. Previous studies have indicated that adequate glucose uptake and normal metabolism were essential for endometrial differentiation and decidualization, providing a nutritional and receptive milieu that supports embryo implantation [22]. The process of endometrial decidualization is inseparable from the activation of glucose metabolism [23]. In RIF patients, glucose transporter 1 expression in endometrial stromal cells is decreased, indicating impaired glucose metabolism [24]. When the embryo is implanted, the endometrium undergoes epithelial-mesenchymal transition. During this stage, the movement of epithelial cells also depends on sufficient energy supply [25]. In addition, progesterone regulates glucose metabolism through the glucose transporter 1, to promote endometrial receptivity. Such receptivity indirectly reflects the impact of energy metabolism/glycolysis on embryo implantation [26]. Obesity has various deleterious effects on human reproduction [27]. Substantial evidence suggests that an increased body mass index, as well as dyslipidemia and LDL-C, not only affect the quality of the oocytes and embryos, but also interfere with embryo implantation and endometrial receptivity, resulting in poor pregnancy outcomes [28,29,30]. As the essential energy sources of the human body, fatty acids contribute to successful embryo implantation. In both animal and human studies, prostaglandin and phospholipid-derived endocannabinoids are two widely studied lipid-derived molecules involved in the establishment of endometrial receptivity [6]. During implantation, prostaglandin levels increase and affect endometrial receptivity through interactions with endometrial epithelial and stromal cells. Compared with fertile controls, prostaglandin synthesis appears to be disrupted in RIF patients [31]. The levels of most phospholipids remarkably increase in the stroma immediately surrounding the implantation sites, uncovering the complexity of the biological processes involved in embryo implantation [32]. Lysophosphatidic acid 3 signaling is a positive factor in embryo implantation and decidualization. RIF patients have decreased levels of LPA3 in the endometrium [31]. Thus, glucose and lipid metabolism imbalance in the endometrium may be a phenotypic alteration that occurs in RIF patients.

Based on the expression profiling of 109 metabolism-related genes, RIF patients were clustered into two subtypes, which had different metabolic characteristics. GSVA analysis further identified that subtype B was more closely associated with lipid metabolism, while subtype A might be related to inflammation pathways. An inflammatory process within the endometrium may cause cellular and biochemical alterations, leading to the risk of RIF. Endometrial inflammation changes the mechanisms securing the timely arrival of a viable blastocyst in a receptive endometrium [33]. Studies have demonstrated that the expression of the genes potentially associated with embryo receptivity and decidualization is likely downregulated in the endometrium in RIF cases with chronic endometritis [34]. On the other hand, embryo attachment-associated inflammation is a balanced and delicately controlled process. Inflammation-prone locations are favorable sites for embryo implantation. In women with RIF, a local biopsy-induced inflammatory response may facilitate the preparation of the endometrium for implantation [35]. Therefore, this new classification helps to accurately determine the RIF etiology and may provide targeted specific intervention strategies to increase pregnancy probability and promote reproductive health.

Then, the eight metabolism-related genes were selected by performing LASSO regression analysis, and by the random forest and the SVM-RFE algorithms. These genes included *SRD5A1*, *POLR3E*, *PPA2*, *PAPSS1*, *PRUNE*, *CA12*, *PDE6D*, and *RBKS*. According to the GSEA analysis, these genes are involved in various metabolic pathways, including amino acid, sugar, and nucleotide sugar metabolism, carbon metabolism, propanoate metabolism, glycolysis/gluconeogenesis, steroid hormone biosynthesis, fatty acid metabolism, and others. They influence the function of the endometrium from different metabolic pathways. A recent study has demonstrated that *SRD5A1* deficiency leads to impaired decidualization, structural and functional changes to decidual blood vessels, and transcriptomic changes affecting angiogenesis signaling pathways [36]. *SRD5A1* silencing has led to a decrease in the progesterone metabolism rate (with higher concentrations of unmetabolized progesterone), indicating that *SRD5A1* plays a crucial role in progesterone metabolism [37]. *SULT1E1* and *PAPSS1* are responsible for estrogen sulfation, by providing enzymes and a universal sulfate donor [38]. The expression levels of *PAPSS1* during the decidualization of human endometrial stromal fibroblast cells were variable [39]. *PPA2* is an important mitochondrial metabolic gene, and its abnormal expression levels can result in mitochondrial dysfunction, leading to embryo implantation failure [40,41]. The expression levels of *CA12* are significantly upregulated in macrophages of human hepatocellular carcinoma, which can enhance the epithelial mesenchymal transition ability of cancer cells and promote tumor metastasis [42]. The changes prepared for implantation in the endometrium include epithelial-mesenchymal transition and proliferation of endometrial cells, suggesting that *CA12* mediated carbon metabolism balance also plays an important role in the formation of endometrial receptivity. 

In this study, the immunological features of RIF and the correlation between eight hub genes and immune cell infiltration levels were analyzed through ssGSEA. As depicted above, the results have shown that most innate and adaptive immune cells presented higher infiltration levels in RIF, compared with control patients. A previous study has indicated that both innate (cytotoxic NK cells, M1 macrophages) and adaptive (Th1 cells, Th17 cells, and B cells) immune cell activation led to embryoblast miscarriage in RIF patients [43]. Abnormal uNK cells may generate adverse outcomes during embryo invasion, such as vascular remodeling, local ischemia, and oxidative stress, which are detrimental to implantation [44]. A recent meta-analysis showed that the proportion of CD56^+^ uNK is significantly increased in RIF patients, when compared with healthy controls [45]. Further experimental research is still needed to elucidate the potential pathophysiology of these immunological characteristic changes.

Furthermore, a risk signature model was established based on gene expression profiles and the coefficients of their association with RIF. Based on this model, the AUC of the ROC curves of the enrolled cohort from our own center was as high as 0.867. With sufficient evidence, we constructed an integrative nomogram to allow for a better prediction of the occurrence of RIF. Last but not least, the eight metabolism-related genes were validated by RT-qPCR in samples from patients from our center. Then, a miRNA-TF-genes network was constructed based on the RegNetwork database, to provide clues for further mechanistic exploration. Further attention to these essential genes may help doctors better manage RIF patients.

Up to now, our study is the first to explore the involvement of metabolism in the pathological mechanisms of RIF. The eight-gene-based metabolism-related characteristic model and the novel classification of RIF, containing two subtypes with different metabolic characteristics, have not been previously published. Nevertheless, in the current study, several limitations should be acknowledged. First, given the complexity and variability of metabolism, larger sample sizes and broader perspectives are needed in order to explore the relations and associations between metabolism and RIF. Second, the sample size of our study was relatively small. Moreover, in vivo and in vitro basic experimental verification is also needed, to elucidate the molecular mechanisms and increase the reliability and accuracy of these results. Additionally, functional validation of the constructed miRNA-mRNA network should be performed in future research.

## 4. Materials and Methods

### 4.1. Data Preprocessing

Raw gene expression data of RIF patients were accessed from the GSE58144 [46], GSE103465 [47] and GSE111974 [48] datasets of the Gene Expression Omnibus (GEO; https://www.ncbi.nlm.nih.gov/gds, accessed on 9 February 2023) database, using the “GEOquery” package [49]. The three datasets were separately based on the platforms of GPL15789 (A-UMCU-HS44K-2.0), GPL16043 [GeneChip^®^ PrimeView™ Human Gene Expression Array (with External spike-in RNAs)] and GPL17077 (Agilent-039494 SurePrint G3 Human GE v2 8x60K Microarray). They contained endometrium samples from 43 RIF patients and 72 healthy individuals, 3 RIF patients and 3 healthy individuals, 24 RIF patients and 24 healthy individuals, respectively. Their expression profiles were incorporated, and batch effects were directly adjusted utilizing the Combat function of the “sva” package [50]. PCA was applied for the dimensional reduction of the transcriptomics data and for evaluating the performance of the Combat function [51].

### 4.2. Differentially Expressed Genes Screening

The “limma” R package [52] was applied to perform DEG screening analysis between the RIF and the healthy control groups. The occurrence of false positives was corrected by the Benjamini-Hochberg multiple-test method. To uncover DEGs, volcano maps and heatmaps were generated using the “ggplot2” and the “pheatmap” packages [53,54]. Statistically significantly upregulated or downregulated genes were used for subsequent analysis.

A total of 1660 human metabolism-related genes (from 86 metabolic pathways) were obtained from the KEGG database (https://www.genome.jp/kegg/, accessed on 9 February 2023) [55]. Differentially expressed metabolism-related genes were identified by intersecting DEGs and metabolic genes and further displayed using the “VennDiagram” package [56].

### 4.3. Molecular Subtypes Identification

Consistency clustering was a resampling-based approach for identifying each member and their subgroup number, as well as validating the cluster. To discover various metabolic patterns, on the basis of those significant metabolism-related DEGs, the “ConsensusClusterPlus” package [57] was implemented.

### 4.4. Functional Enrichment Analysis

To explore the biological significance of DEGs, we performed Gene Ontology (GO) classification [58] and KEGG pathway analysis using the “clusterProfiler” package [59]. Furthermore, to explore the differences in biological processes between the different subgroups, the KEGG and Reactome gene sets were downloaded from the Molecular Signature Database (http://software.broadinstitute.org/gsea/msigdb, accessed on 9 February 2023) as the reference set [60]. GSVA was performed to demonstrate the signaling pathways alteration between the two clusters using the “GSVA” R package [61]. GSEA was implemented for functionally elucidating the biological significance of characteristic genes. For achieving a normalized enrichment score for each analysis, gene set permutations with 1000 times were conducted. Only terms with a false discovery rate < 0.05 were considered as statistically significant enrichment.

### 4.5. Characteristic Gene Selection 

Three machine learning algorithms, LASSO [62], random forest [63] and SVM-RFE [64], were employed for screening key genes. We used the “randomForest” package [65] for random forest and the “glmnet” package [66] to perform LASSO logistic regression with a turning/penalty parameter utilizing a 10-fold cross-verification. The SVM classifier was created using the R package “kernlab” [67]. The three aforementioned classifiable models’ overlapping genes were then uncovered. 

### 4.6. Receiver Operator Characteristic Analysis and Nomogram Construction

Subsequently, the ROC curves were plotted, and the AUC were separately calculated to evaluate the performance of each signature, using the “rms” and “ROCR” packages [68,69]. Next, characteristic genes were incorporated to establish a nomogram using logistic regression analysis. The calibration curve was utilized for evaluating the accuracy of the nomogram. The clinical usefulness of the nomogram was assessed through decision curve analysis.

### 4.7. Patient Recruitment for External Validation

Human endometrial tissues were collected from women attending the Department of Obstetrics and Gynecology, PKUPH, China. All samples were surplus tissue from endometrial biopsies obtained from patients for diagnostic purposes, between January 2018 and December 2022. The timing of the endometrial biopsy was 5 days after ovulation (evaluated by ultrasonography), which is equivalent to LH + 7 in a natural cycle. The patients in the RIF group had suffered at least three embryo transfer failures, in which at least four morphologically high-grade embryos were transferred in total. There were no other obvious explanations for the occurrence of RIF. The control group included women who received artificial reproductive technology due to obstruction of the fallopian tube or male infertility and confirmed their ability to conceive. 

The total RNA was extracted from patient tissue with TRIzol^®^ (Tiangen Biotech Co., Ltd., Beijing, China) and reverse transcribed into cDNA using the FastQuant RT Kit (Tiangen Biotech Co., Ltd., Beijing, China). RT-qPCR was performed on a MiniOpticon Real-Time PCR #CFB3120EDU (Bio-Rad, Hercules, CA, USA) machine, using the SuperReal PreMix Plus reaction mixture (SYBR Green) (Tiangen Biotech Co., Ltd., Beijing, China). Gene expression levels relative to GAPDH expression levels were assessed using the 2^−ΔΔCt^ method. Experiments were conducted in triplicate. Primer sequences (Sangon Biotech, Shanghai, China) for the reference and candidate genes are listed in Table 2.

### 4.8. Immune Cell Infiltration Evaluation

Single-sample gene set enrichment analysis (ssGSEA) of the R package “GSVA” was utilized to evaluate the infiltration levels of immune cells in individuals with different metabolic patterns [70].

### 4.9. Metabolism-Related Transcription Factor/miRNA Regulatory Network Construction

To study the regulatory mechanisms related to the eight crucial genes, TFs, and miRNAs that bind to hub genes, the RegNetwork target gene prediction database (http://www.regnetworkweb.org/, accessed on 12 February 2023) was used. A TF-miRNA-gene regulatory network was constructed and visualized using Cytoscape (Version 3.9.1). 

### 4.10. Statistical Analysis

Statistical analyses were performed using R software (Version 4.2.2). Continuous variables were presented as the mean ± SD (standardized deviation). Categorical variables were described by frequency (n) and proportion (%). Statistically significant differences among variables were assessed using Student’s *t*-tests, nonparametric tests, Chi-square tests, or one-way ANOVA tests. The correlation between the variables was determined using Pearson’s or Spearman’s correlation test. All statistical tests were two-tailed, and a *p* value threshold of 0.05 was considered statistically significant.

## 5. Conclusions

In summary, our study investigated the specific metabolism-related molecular mechanisms of RIF and determined eight characteristic metabolism-related genes (*SRD5A1*, *POLR3E*, *PPA2*, *PAPSS1*, *PRUNE*, *CA12*, *PDE6D* and *RBKS*) that could possibly predict the occurrence of RIF. Moreover, we proposed a new molecular classification comprising two RIF subtypes with different metabolic characteristics. Thus, our study may provide new insights into the pathogenesis of metabolic disorders affecting human reproductive health.

## Figures and Tables

**Figure 1 ijms-24-13488-f001:**
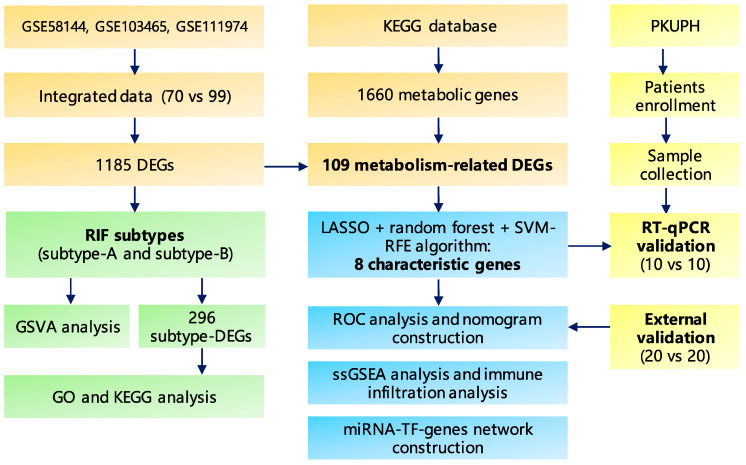
Diagram of the study design. Abbreviations: DEGs: differently expressed gene, GO: Gene Oncology, ssGSEA: single-sample gene set enrichment analysis, GSVA: Gene set variation analysis, KEGG: Kyoto Encyclopedia of Genes and Genomes, LASSO: least absolute shrinkage and selection operator, miRNA: microRNA, PKUPH: Peking University People’s Hospital, RIF: recurrent implantation failure, ROC: receiver operating characteristic, RT-qPCR: real time-quantitative PCR, SVM-RFE: support vector machine-recursive feature elimination, TF: transcription factor.

**Figure 2 ijms-24-13488-f002:**
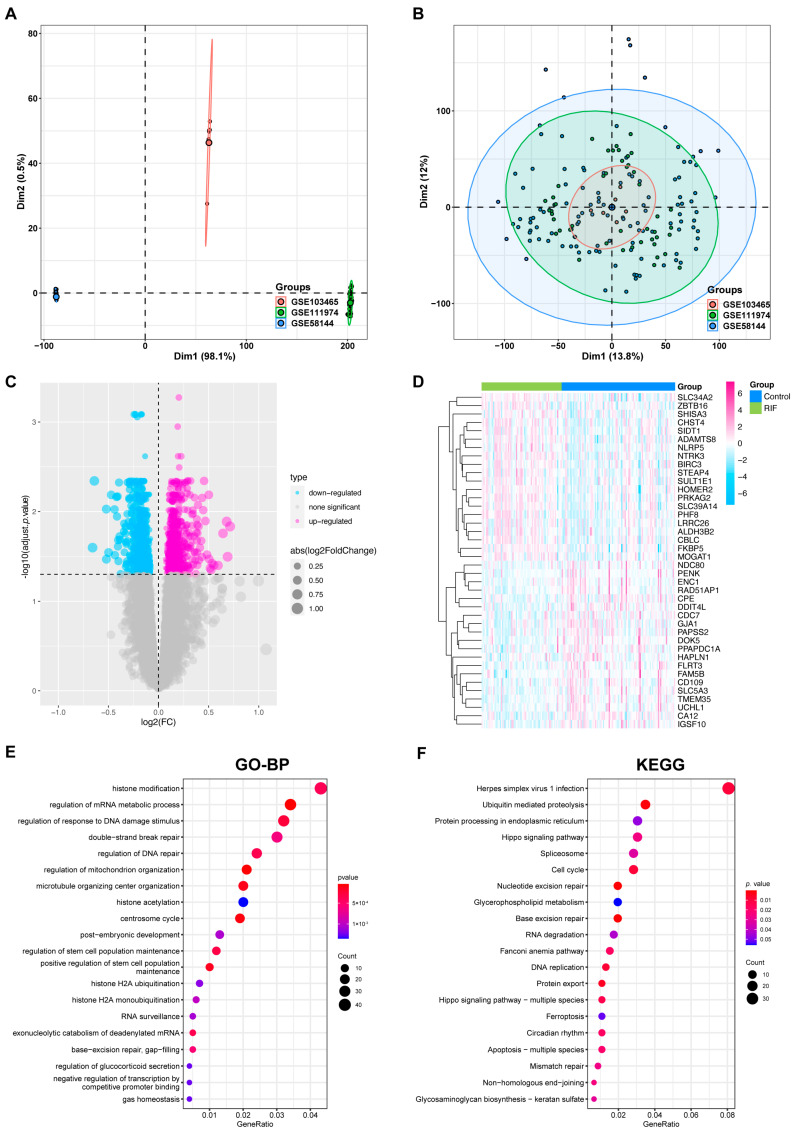
Data processing and DEGs identified of the derivation cohort. PCA of GSE58144, GSE103465 and GSE111974 datasets before (**A**) and after (**B**) batch correlation. (**C**) RIF-related DEGs volcano plot with log2FoldChange in the horizontal coordinate and −log10 (adjust *p* value) in the vertical coordinate. (**D**) Heatmap of RIF-related DEG expression levels: pink indicates high gene expression, and light blue indicates low gene expression. (**E**) Main BPs and (**F**) KEGG pathways enriched by RIF-related DEGs. Each term’s *p* value is colored according to the legend. Abbreviations: BP: biological process, DEG: differentially expressed gene, GO: Gene Oncology, KEGG: Kyoto Encyclopedia of Genes and Genomes, PCA: principal component analysis, RIF: recurrent implantation failure.

**Figure 3 ijms-24-13488-f003:**
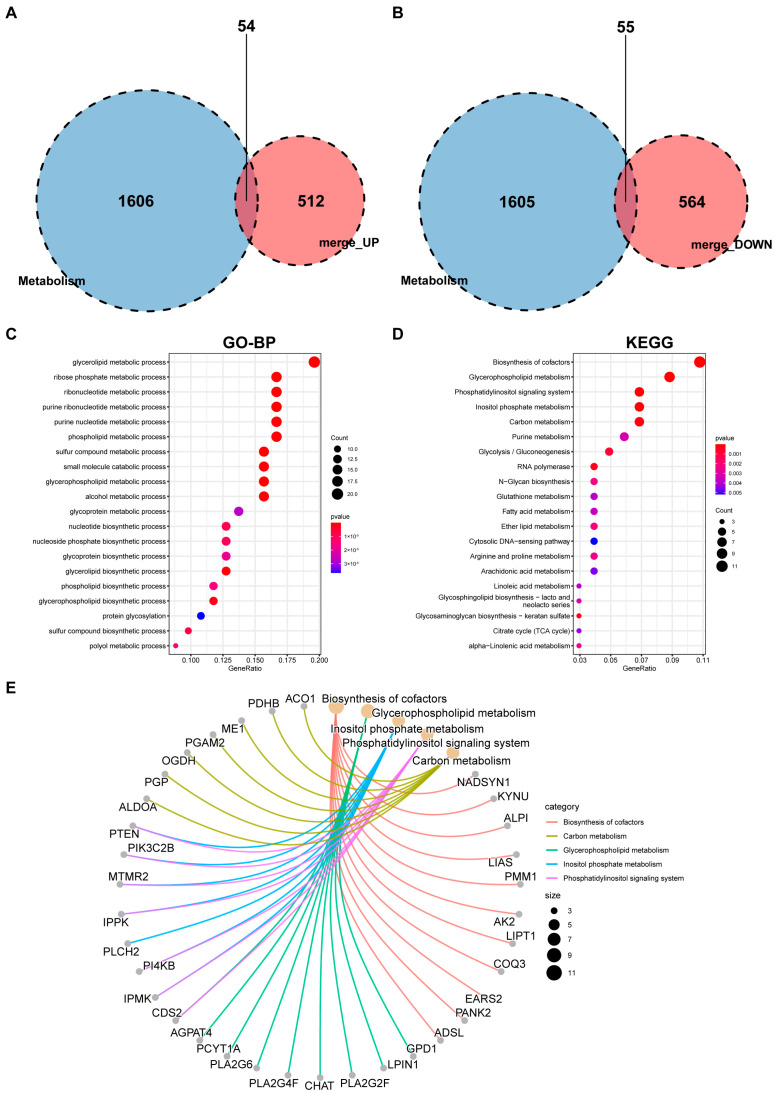
Screening of metabolism-related DEGs and enriched items in GO and KEGG analyses. Venn diagram of (**A**) metabolic genes and upregulated DEGs, and (**B**) metabolic genes and downregulated DEGs. (**C**) Enriched items in GO-BP analysis. (**D**) Enriched items in KEGG pathway analysis. Each term’s *p* value is colored according to the legend. Different colored bubbles reflect different pathway terms. (**E**) Net plot showing the top 5 signaling pathways enriched by KEGG analysis. Abbreviations: DEG: differentially expressed gene, GO: Gene Ontology, BP: biological process, KEGG: Kyoto Encyclopedia of Genes and Genomes.

**Figure 4 ijms-24-13488-f004:**
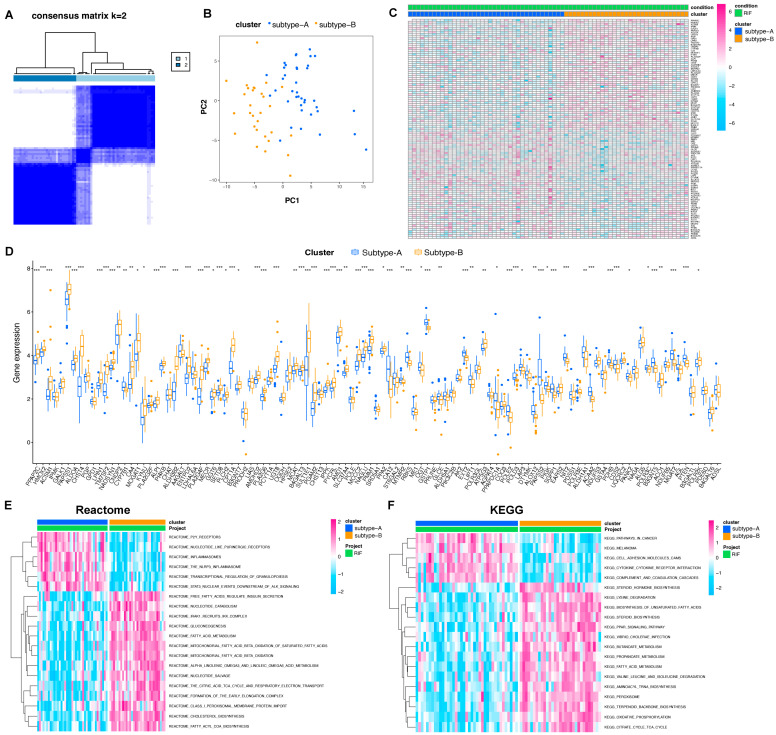
Construction of two metabolic subtypes of RIF based on metabolism-related DEGs. (**A**) Consensus matrix heatmap when K = 2. (**B**) PCA plots demonstrating that RIF specimens are categorized as two subtypes (subtype-A and subtype-B) in accordance with the expression profiling of metabolism-related DEGs. (**C**) Heatmap visualizing the expression of metabolism-related genes in the two subgroups. (**D**) Box plots showing the mRNA expression of characteristic genes in two metabolic subtypes. * *p* < 0.05; ** *p* < 0.01; and *** *p* < 0.001. (**E**) Reactome and (**F**) KEGG terms are utilized for GSVA illustrating the difference in metabolic pathways between two subtypes. Abbreviations: DEG: differentially expressed gene, GSVA: gene set variation analysis, KEGG: Kyoto Encyclopedia of Genes and Genomes. PCA: principal component analysis, RIF: recurrent implantation failure.

**Figure 5 ijms-24-13488-f005:**
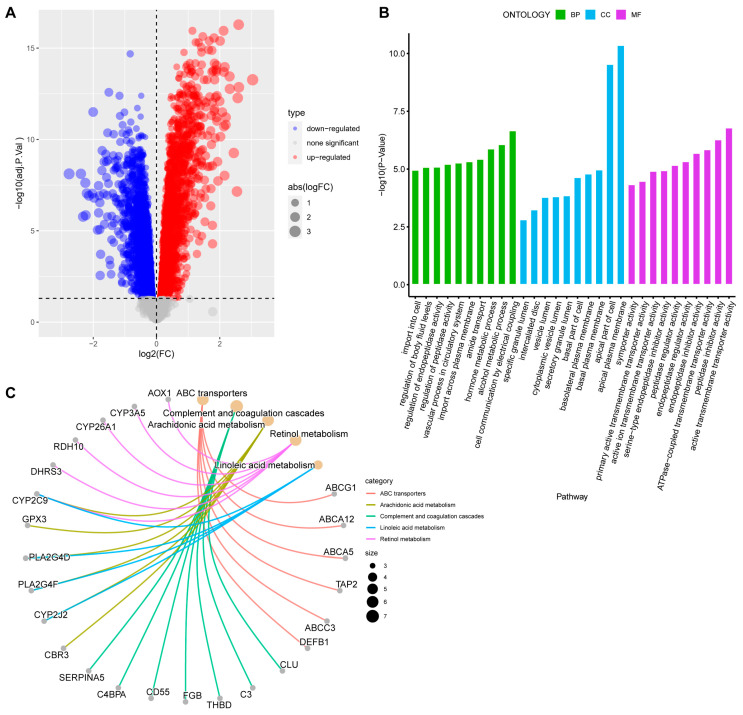
DEGs analysis of two subtypes and functional analyses. (**A**) Volcano plot showing the DEGs between the two subgroups. (**B**) Bar plot visualizing the biological processes enriched by GO analysis. (**C**) Net plot showing the top 5 signaling pathways enriched by KEGG analysis. Abbreviations: DEG: differentially expressed genes, BP: biological process, CC: cellular component, MF: molecular function, GO: Gene Ontology, KEGG: Kyoto Encyclopedia of Genes and Genomes.

**Figure 6 ijms-24-13488-f006:**
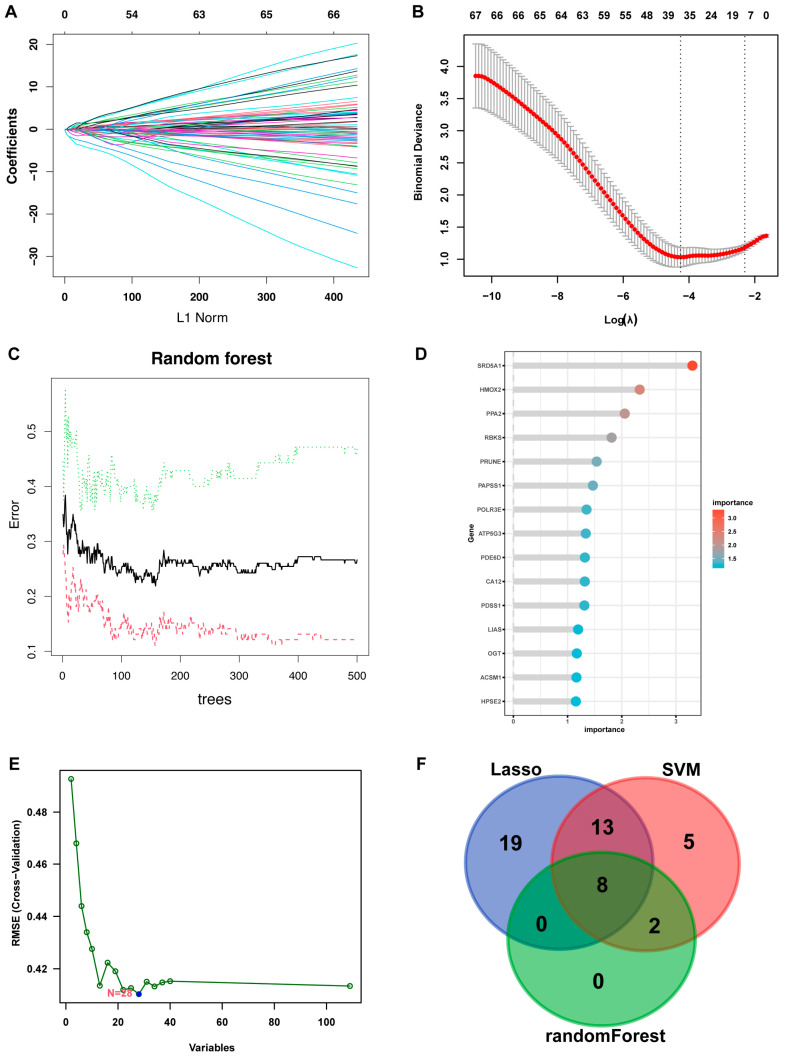
Selection of characteristic genes among key metabolism-related DEGs. (**A**) LASSO logistic regression algorithm to screen associated genes with ten-time cross-verification. Each curve corresponds to a single gene. (**B**) LASSO coefficient profiling. (**C**) Random forest for the relationships between the number of trees and error rate. (**D**) The rank of genes in accordance with their relative importance. (**E**) SVM-RFE algorithm for feature selection. (**F**) Venn diagram showing the overlapping genes. Abbreviations: LASSO: least absolute shrinkage and selection operator, SVM-RFE: support vector machine-recursive feature elimination.

**Figure 7 ijms-24-13488-f007:**
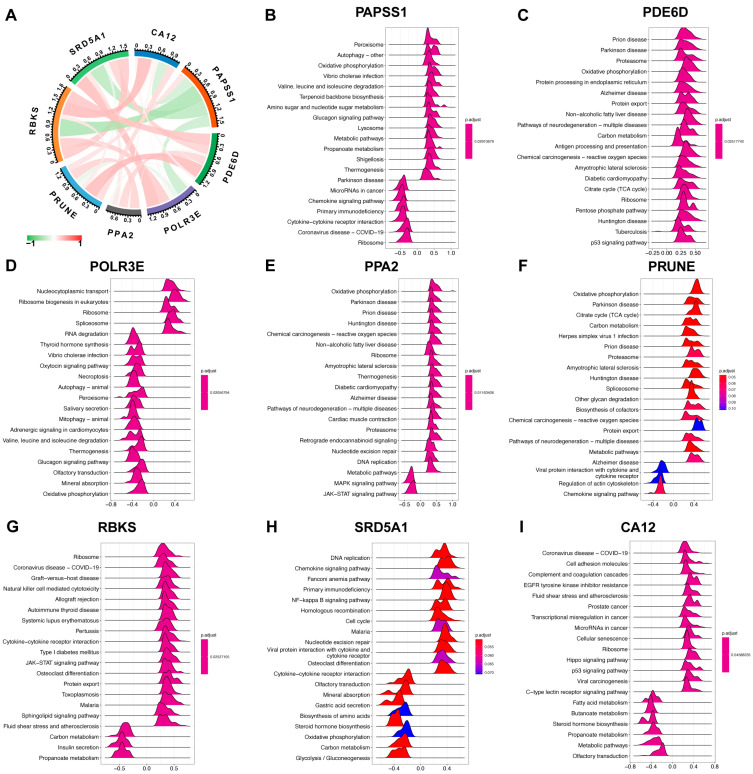
The correlation analysis and GSEA signaling pathways involved in the characteristic genes. (**A**) Interactions between characteristic genes at the molecular level. The red line represents positive correlation, the green line represents negative correlation, and the darker the color, the stronger the correlation. (**B**–**I**) The main signaling pathways that are significantly enriched in high expressions of characteristic genes. The x-axis displays the enrichment fractions, and greater than 0 indicates a positive correlation between genes and pathways, and less than 0 indicates a negative correlation. Abbreviations: GSEA: gene set enrichment analysis.

**Figure 8 ijms-24-13488-f008:**
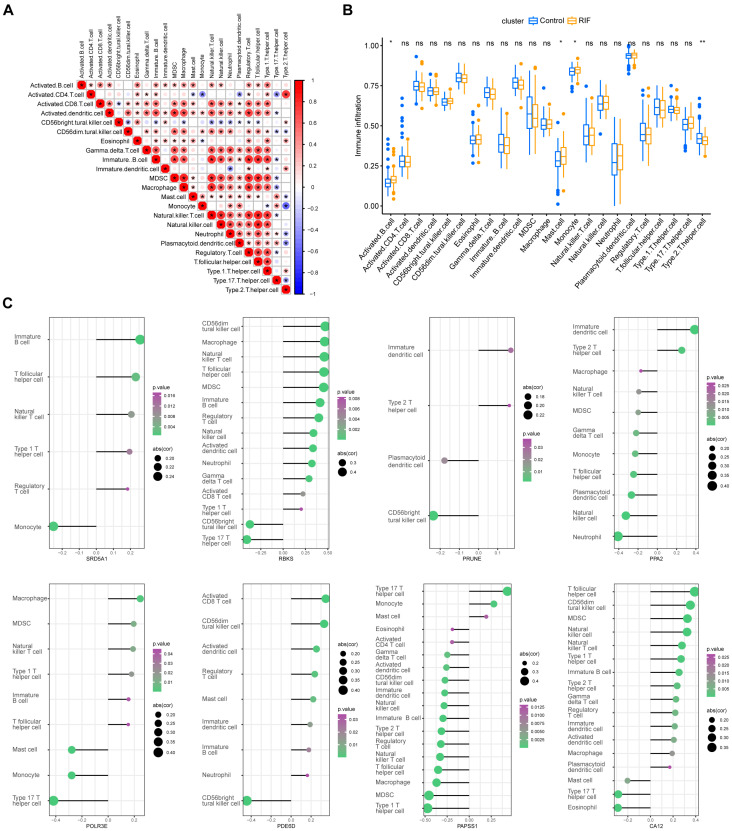
Immunological infiltration features of RIF. (**A**) Heatmaps depicting the correlations between distinct immune cell compositions in RIF. The size of the colored bubbles represents the strength of correlation. The red bubble represents positive correlation, the blue bubble represents negative correlation, and the bigger and darker the color, the stronger the correlation. (**B**) Box plots depicting the infiltration levels of immune cells in RIF (orange) and normal (blue) tissues. ns *p* ≥ 0.05, * *p* < 0.05, and ** *p* < 0.01. (**C**) Correlation between each characteristic gene and infiltrating immune cells. The size of the dots represents the strength of the correlation between genes and immune cells; the larger the dots, the stronger the correlation. The color of the dots represents the *p* value. Abbreviations: RIF: recurrent implantation failure.

**Figure 9 ijms-24-13488-f009:**
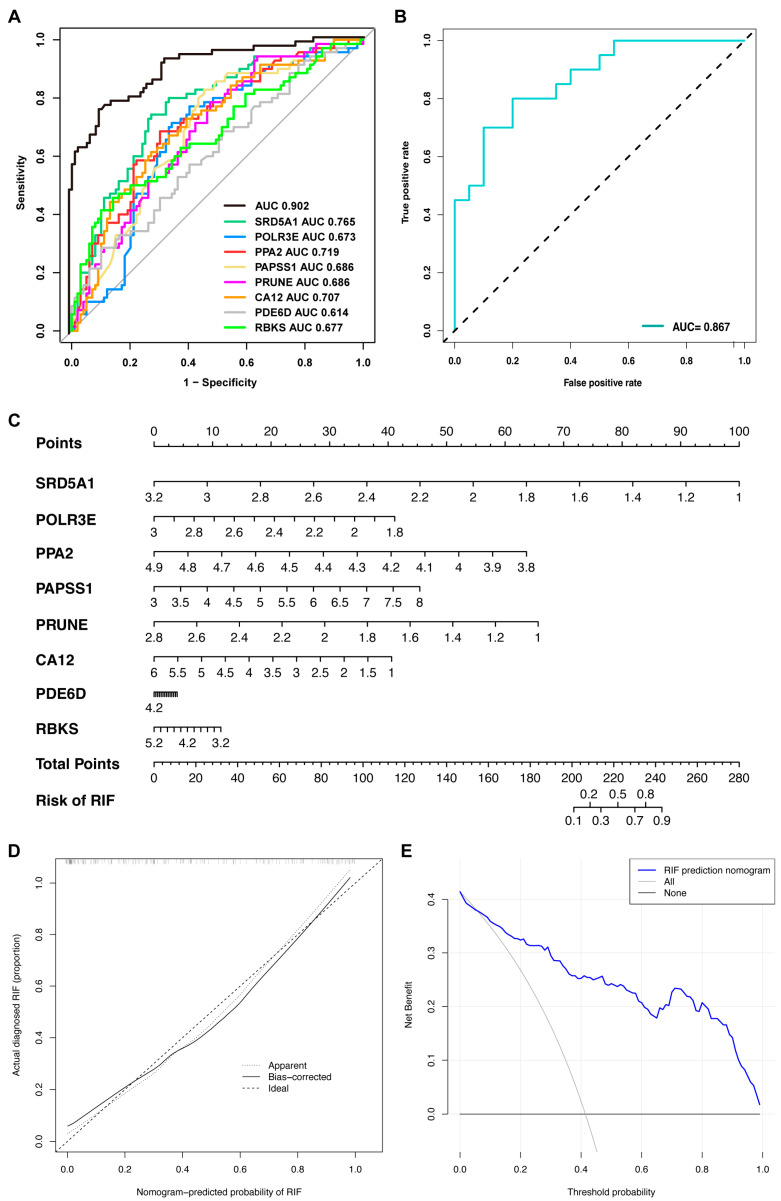
Diagnostic efficacy and validation of characteristic genes in predicting RIF. The ROC curves estimating the diagnostic performance of characteristic genes (**A**) in the combined GSE58144, GSE103465 and GSE111974 datasets (AUC = 0.902), and (**B**) in the validated cohort patients enrolled from PKUPH (AUC = 0.867). (**C**) Establishment of a nomogram integrating characteristic genes for predicting RIF. In the nomogram, each variable corresponds to a score, and the total score can be calculated by adding the scores for all variables. (**D**) Calibration curve estimates the prediction accuracy of the nomogram. (**E**) Decision curve analysis shows the clinical benefit of the nomogram. Abbreviations: AUC: area under curve, RIF: recurrent implantation failure, PKUPH: Peking University People’s Hospital, ROC: receiver operating characteristic.

**Figure 10 ijms-24-13488-f010:**
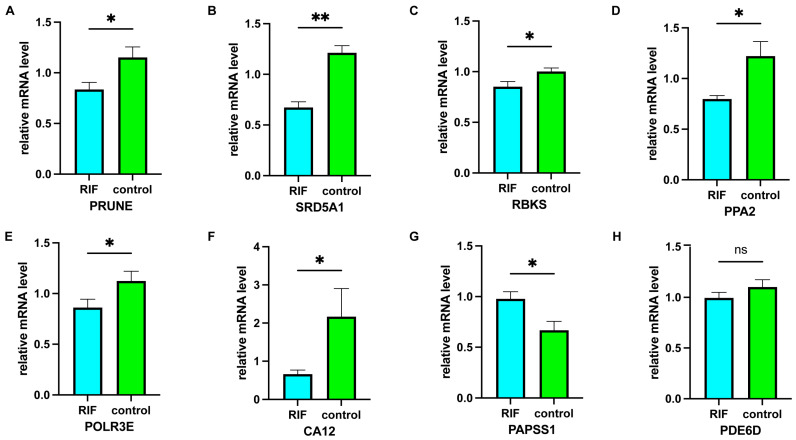
RT-qPCR validation in the endometrium of days 20–24 in the luteal phase of the menstrual cycle in 20 patients enrolled from PKUPH. the mRNA expression levels of (**A**) *PRUNE*, (**B**) *SRD5A1*, (**C**) *RBKS*, (**D**) *PPA2*, (**E**) *POLR3E*, (**F**) *CA12* were significantly lower, and the mRNA expression levels of (**G**) *PAPSS1* were significantly higher in the RIF group. The mRNA expression of (**H**) *PDE6D* in the RIF group was lower, but there was no statistical difference between the two groups. ns *p* ≥ 0.05, * *p* < 0.05; ** *p* < 0.01. Abbreviations: PKUPH: Peking University People’s Hospital, RIF: recurrent implantation failure, RT-qPCR: real time-quantitative PCR.

**Figure 11 ijms-24-13488-f011:**
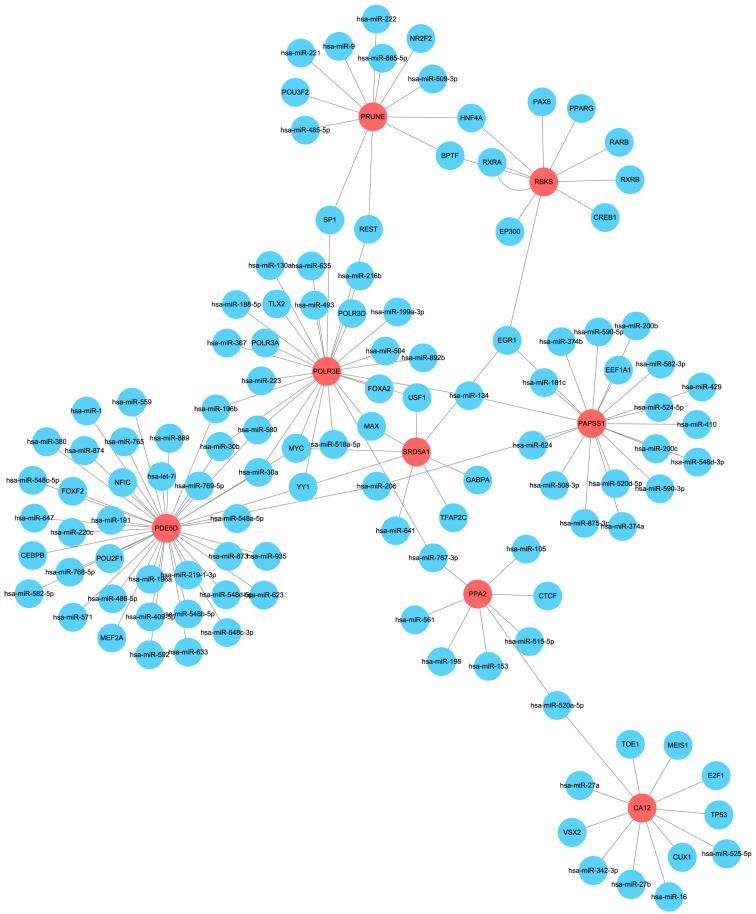
The miRNAs-TFs-genes networks. Abbreviations: miRNA: microRNA; TFs: transcription factor.

**Table 1 ijms-24-13488-t001:** Characteristics of patients in validation cohort from Peking University People’s Hospital.

Clinical Parameter	Control N = 20	RIF N = 20	*p* Value
Age (years)	34.00 ± 3.34	36.85 ± 2.46	0.004
BMI (kg/m^2^)	20.10 ± 5.17	21.90 ± 5.79	0.306
Infertility duration (years)	3.11 ± 1.76	4.30 ± 2.23	0.072
Infertility			0.507
Primary infertility	14 (70.00%)	12 (60.00%)	
Secondary infertility	6 (30.00%)	8 (40.00%)	
Basal FSH (IU/L)	8.75 ± 1.77	8.50 ± 5.12	0.838
Basal LH (IU/L)	4.21 ± 1.81	3.40 ± 1.88	0.179
Prolactin (pg/mL)	12.37 ± 5.50	12.53 ± 6.44	0.936
Basal estradiol (pg/mL)	38.70 ± 13.11	67.76 ± 117.02	0.277
Androgen (pg/mL)	1.84 ± 0.60	1.73 ± 0.96	0.689
AMH (ng/mL)	3.55 ± 2.50	5.32 ± 7.08	0.301
AFC	6.67 ± 2.64	5.08 ± 2.76	0.083
Endometrial type			0.885
Type-A	11 (64.71%)	9 (64.29%)	
Type-B	4 (23.53%)	4 (28.57%)	
Type-C	2 (11.76%)	1 (7.14%)	

Abbreviations: AFC, antral follicle count; AMH, anti-Müllerian hormone; BMI, body mass index; FSH, follicle-stimulating hormone; LH, luteinizing hormone; RIF, recurrent implantation failure.

**Table 2 ijms-24-13488-t002:** PCR primers.

Gene	Forward Primer Sequence	Reverse Primer Sequence
PRUNE	CTTGAAGATAGGCATGGAGGTTAGG	CAACGATCTGTGAAGTCCTGGAAC
SRD5A1	CCTGCCGCTCTACCAGTACG	TCCTCCTCGCATCAGAAATGGG
RBKS	GAAGCAGTTCCTGTAGCAGCATC	TGGTGTGTAAGGTTGGCAAAGATTC
PPA2	AAGGGAAGATATTCGCCACATAGC	GCCACCAAGGAGCCAATGAATC
PDE6D	TGAACCTTCGGGATGCTGAGAC	CCACACCAGGGACAGACAGG
PAPSS1	AGCAACCAATGTCACCTACCAAG	CAACCACGAAAGCCACCTCTG
CA12	TCTTGGTGGCTGGCTTGTAAATG	CATCTGTATTGTGGTGGTGGTGTC
POLR3E	GCCAACTTGATGAGCCTCCTG	GACCAACATCGCCACCTTCTG

## Data Availability

The datasets used and/or analyzed in the current study are available in the GEO database (https://www.ncbi.nlm.nih.gov/gds, accessed on 12 July 2023). Further inquiries can be directed to the corresponding authors.

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
