# Peer review of "Recurrent Implantation Failure: Bioinformatic Discovery of Biomarkers and Identification of Metabolic Subtypes"

_ijms, 2023, doi:10.3390/ijms241713488_

Round 1

Reviewer 1 Report

This is an interesting paper focused on few aspects of embryos implantation into the prepared endometrium With specific approach focusing 8 promising-related genetic biomarkers and 2 metabolic -related subtypes of RIF patients. The main focus of this study was the endometrial metabolic and immune features of RIF patients to orient the future investigations

A.      The implantation of human embryo is a complex process with a wide range of factors concurring to success and unfortunately not related one each other. Since more than 30 years ago various front of researchers studied the coordination between embryo and endometrial differentiation to gain the factor or factors promoting the embryo implantation without success. The most promising study was that of Stewart on Leukaemia Inhibitory Factor around 90’ that was the key factor in the mice but not in humans.

B.     I discourage the authors to emphasize the term window of implantation in considering the adequacy of the endometrial features to embryo implantation because only the successful implantation characterize this term not any other characterization. Neither the gene sequence features already published and commercialized because they were recently established not usable for prognostic value

C.     In the recent systematic review ( ESHRE) the role of immunological cell phenotypes within the endometrium were not considered discriminating endometrial feature adequacy versus non adequate

D.    The intimate role of embryo quality in the implantation destiny was already investigated and represent the major determinant . In close relationship with age ( Ata B et al 2021) . And with the age independently from the embryionic genetic ( Vitagliano A et al 2023) . Thus here we are talking about possible feature imprinting indefinite amount of influence on the entire process

E.     The other factors influencing the successful implantation are mostly identified but are not still quantified . Part of that factors are genomic , other non genomic ( e g Uterine contractility) The transcriptomic factor for immunology and metabolism are two of them.

F.     Authors to predict the prevalence of RIF in patients, created a  diagnostic nomogram was based on the eight characteristic genes   emphasizing that a decision-making based on their line graph model could be beneficial for RIF patients  : the robust clinical validation is required before to make any conclusion. The prognostic value  of whatever test performed today and valid tomorrow on the unique human tissue undergo cyclically shedding and rebuilding is something to strongly demonstrate . Otherwise it is a speculation

In summary, this study investigated the specific metabolism associated ( related can be used only when robust clinical validation should be performed)  with some molecular mechanisms observed of RIF, and the authors determined eight characteristic metabolism-related genes (SRD5A1, 551 POLR3E, PPA2, PAPSS1, PRUNE, CA12, PDE6D and RBKS) that they propose to predict the occurrence of part  of RIF.  

After my review I confirm that this study is an interesting study and that could be of interest for the readers of the journal I suggest the authors to modify slowly the absolute affirmer terms used for its results and application in more relative an suborder to further clinical robust study to validate these remarks.  Limiting their ambitions

Reviewer 2 Report

Type of article: research article

Title of article: Recurrent implantation failure: bioinformatic discovery of biomarkers and identification of metabolic subtypes

The manuscript is well-written and offers a thoughtful interpretation of the obtained data. Overall, a good and detailed work, the result of a careful predominantly (but extensive) in silico investigation adequately represents the subject matter. The data discussed are analyzed objectively by placing the necessary criticisms. The discussion is well structured with an accurate analysis of the obtained data. The topic of the work is in agreement with the journal aim. The experimental design is well performed. One of the strengths of the work is the huge number of high throughput data analyzed, as well as the different analyses performed. The main data of the work provide the identification of eight promising metabolism-related genetic biomarkers, namely SRD5A1, POLR3E, PPA2, PAPSS1, PRUNE, CA12, PDE6D, and RBKS, as well as two metabolic-related subtypes of recurrent implantation failure patients. Figures should be improved given the low resolution, while table 1 is highly informative and clear.

Figures should be improved given the low resolution. The majority of figures are fundamentally unreadable. Moreover, several panels are too small and words impossible to be read. The work cannot be accepted without strong resolution improvement of the figures.

In line 14, it should be “analyses”

Please include more literature on the molecular basis and mechanisms of recurrent implantation failure: DOI: 10.1080/14647273.2021.1956693, DOI: 10.1080/19396368.2017.1310329, https://rbej.biomedcentral.com/articles/10.1186/s12958-021-00778-1, https://www.frontiersin.org/articles/10.3389/fendo.2022.1061766/full , doi: 10.3390/ijms221810082 

Why don’t include in the abstract the number of analyzed samples? The number of patients from which data were retrieved for enrichment analyses as well as the number of patients and controls for data validation. The same observation can be made for the study workflow in figure 1. 

Impairments in the maternal immune system during pregnancy can improve the susceptibility to viral infections, ultimately leading to an impairment of the embryo formation (authors can check/use DOI:10.3389/fmicb.2021.789991 and doi: 10.1155/2013/752852). This information should be included in lines 63-70 

Have enrichment analyses been performed in separated upregulated and downregulated DEGs? This gene stratification might be more informative on the expressed genes in identified enriched pathways

Concerning GO analyses of figure 3, besides GO for biological function, have GO for cellular component and molecular functions analyses been performed?

If possible, please avoid abbreviations in the subhead titles. For instance, line 140

In the methods, given the lack of supporting reference, this recently published work on the PCA analysis (doi:10.1002/jmv.28949) should be included in lines 459-460

Unless differently indicated, I suggest moving conclusions after the discussion

Case and control characteristics should be statistically compared and p values depicted in table 1. 

An additional limitation in lines 437-445 is the lack of functional validation of the constructed miRNA-mRNA network. Was some of these interactions validated by others?
